# Impartial Selection with Predictions[*]

**Javier Cembrano**
Department of Algorithms and Complexity
Max-Planck-Institut für Informatik
Saarbrücken, Germany
Department of Industrial Engineering
Universidad de Chile
Santiago, Chile
`jcembran@mpi-inf.mpg.de`

**Felix Fischer**
School of Mathematical Sciences
Queen Mary University of London
London, UK
`felix.fischer@qmul.ac.uk`

**Max Klimm**
Institute of Mathematics
Technische Universität Berlin
Berlin, Germany
`klimm@tu-berlin.de`

## Abstract

We study the selection of agents based on mutual nominations, a theoretical problem with many applications from committee selection to AI alignment. As agents both select and are selected, they may be incentivized to misrepresent their true opinion about the eligibility of others to influence their own chances of selection. Impartial mechanisms circumvent this issue by guaranteeing that the selection of an agent is independent of the nominations cast by that agent. Previous research has established strong bounds on the performance of impartial mechanisms, measured by their ability to approximate the number of nominations for the most highly nominated agents. We study to what extent the performance of impartial mechanisms can be improved if they are given a prediction of a set of agents receiving a maximum number of nominations. Specifically, we provide bounds on the consistency and robustness of such mechanisms, where consistency measures the performance of the mechanisms when the prediction is accurate and robustness its performance when the prediction is inaccurate. For the general setting where up to $k$ agents are to be selected and agents nominate any number of other agents, we give a mechanism with consistency $1 - O\left(\frac{1}{k}\right)$ and robustness $1 - \frac{1}{e} - O\left(\frac{1}{k}\right)$. For the special case of selecting a single agent based on a single nomination per agent, we prove that 1-consistency can be achieved while guaranteeing $\frac{1}{2}$-robustness. A close comparison with previous results shows that (asymptotically) optimal consistency can be achieved with little to no sacrifice in terms of robustness.

## 1  Introduction

Majority voting is a simple but very important mechanism for collective decision making. Its use dates back at least to ancient Athens, where it was employed for example to decide on the expulsion of citizens from the city [18]. A much more recent proposal uses majority voting to aggregate the solutions of multiple calls to large language models (LLMs) [14]. Some proposals even go so far as

---

[*]The full version of the paper can be accessed on arXiv:2510.19002.

39th Conference on Neural Information Processing Systems (NeurIPS 2025).

using it in AI alignment, and destroying AI entities if they are perceived as unaligned with human ethics by other AI entities [24]; see also Aaronson [1], Irving et al. [20]. We may formalize this idea by considering a fixed number of different AI entities that can nominate other entities for being incompatible with human values. The entity that receives the most nominations in that way would then be destroyed.

The motivation for using majority decisions in these applications is their superior robustness to outliers compared to decisions made by a single entity. This argument requires, of course, that each entity is incentivized to reveal its true opinion about others rather than following its selfish interests. This is true for voting in general, but even more so in settings like those described above where the set of candidates and the set of voters overlap or are the same. Indeed, it is reasonable to assume that an Athenian citizen in fear of expulsion would have cast their vote for someone they considered likely to receive a large number of nominations, rather than someone they considered worthy of expulsion, in order to minimize their own risk of being expelled. Similarly, it is naïve to assume that AI entities risking destruction due to misalignment will truthfully report on the misalignment of other entities if this negatively affects their own chances of survival. What is needed are voting mechanisms for which the probability that an entity is selected is independent of the nominations cast by that entity. Such mechanisms are called *impartial* in the literature.

While impartiality is obviously appealing, previous work has established strong impossibility results for mechanisms that satisfy it. Deterministic impartial mechanisms that select a fixed number $k$ of entities must fail natural axioms [12, 19], and the overall number of nominations for the selected entities cannot provide a constant approximation to the maximum number of nominations for any set of $k$ entities [3]. Even randomized impartial mechanisms are relatively limited; for example, for the selection of a single entity they can only approximate the maximum number of nominations to a factor of $\frac{1}{2}$ [3, 17].

To improve the performance of impartial mechanisms, we will assume that the mechanism has access to a *prediction* of the entities most suitable for selection. Depending on the application, the prediction could for example come from another LLM not participating in the voting process or from expert advice. The prediction should not be thought of as a prediction about how the votes will turn out, but rather about who is most appropriate for selection. In particular, the prediction is independent of the votes and, thus, following the prediction is impartial. This assumption, while being crucial for our analysis, is satisfied in many scenarios. For a concrete example, consider a situation where a group of agents cast votes on one another about who should be considered for a promotion. The group feeds all CVs to an LLM, asking it for its opinion on who is the best candidate and taking its output as a prediction. In a second step, they can use one of the mechanisms in this paper to do a formal vote. This two-step process has the advantage that when the output of the LLM does not align at all with the opinions of the group, they can overrule its decision. In addition, the process is impartial in the sense that nobody can influence their own chance of being promoted. The guarantee of impartiality applies regardless of whether the agents know the prediction before casting their votes or not.

Our work is part of a growing literature on algorithms and mechanisms with advice; a website maintained by Lindermayr and Megow [22] provides an excellent overview of the area. The area is motivated by the fact that LLMs often provide astonishingly accurate answers, but also sometimes fail spectacularly. Mechanisms with advice therefore need to be able to cope with good as well as bad predictions, without a clear way to distinguish between the two. This trade-off is studied formally by considering the *consistency* and *robustness* of a mechanism. The consistency of a mechanism describes its ability to produce good outcomes when the predictions are accurate; the robustness its ability to produce reasonable results even when the predictions are inaccurate. The ability of a mechanism to move gracefully between these extremes is referred to as *smoothness*.

We will specifically consider deterministic and randomized impartial selection mechanisms with predictions. As it is standard in the literature on impartial selection, we formalize nominations among entities as a directed graph, where the set $[n] = \{1, \ldots, n\}$ of vertices represent the entities and an edge from $i$ to $j$ indicates that $i$ casts a nomination for $j$. A deterministic $k$-selection mechanism with predictions is given such a graph and a prediction $\hat{S} \subseteq [n]$ with $|\hat{S}| = k$, and returns a set of at most $k$ vertices. A randomized $k$-selection mechanism is a lottery over deterministic mechanisms. Letting $\Delta_k$ denote the maximum sum of indegrees of any $k$ vertices in the graph, a mechanism is called $\alpha$-consistent for some $\alpha \in [0, 1]$ if the (expected) sum of indegrees of the selected vertices is at least $\alpha \Delta_k$ when the prediction is accurate, i.e., when the total indegree of the vertices in $\hat{S}$

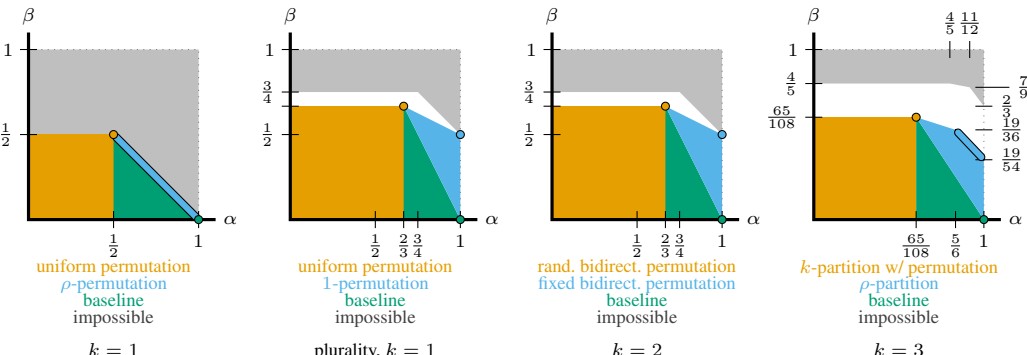

Figure 1: Trade-off between $\alpha$-consistency and $\beta$-robustness of impartial $k$-selection mechanisms. Orange dots are the best mechanisms from previous work; orange areas are the whole ranges of possible consistency–robustness combinations implied by them. Green dots are the trivial mechanisms always selecting the predicted set; green areas are new ranges of consistency–robustness combinations implied by lotteries of them with previous work. Blue dots and blue rounded rectangles are new mechanisms introduced in this paper; blue areas are new ranges of consistency–robustness combinations implied by them or by lotteries of them with previous work. Gray areas are impossible consistency–robustness combinations as shown in Theorem 6.1. Whether the combinations in the white areas are achievable by impartial mechanisms is left for future research.

is indeed equal to $\Delta_k$. While it is trivial to achieve 1-consistency in an impartial way, by simply returning the predicted set $\hat{S}$, this would lead to arbitrarily bad performance when the predictions are inaccurate. To measure the performance of a mechanism in such cases, a mechanism is called $\beta$-robust for some $\beta \in [0, 1]$ if the (expected) sum of indegrees of the selected vertices is at least $\beta\Delta_k$ regardless of the quality of the prediction. We will be interested in the largest possible values of $\alpha$ and $\beta$ for which impartial $\alpha$-consistent and $\beta$-robust mechanisms can be found.

**Our Results.** We study impartial mechanisms with predictions in different settings; Figure 1 summarizes our results and compares them with previous work. As we initiate the study of impartial mechanisms with predictions, all previous mechanisms are unable to deal with predictions and consequently have equal robustness and consistency. Comparing our results with the baseline mechanism defined as a lottery between the best known mechanisms from the literature and the trivial mechanism that always selects the predicted set shows significant improvements.

We first study the classic setting of randomized impartial 1-selection mechanisms for approval voting. We propose a family of mechanisms we call $\rho$-permutation mechanisms that are parametrized by a confidence parameter $\rho \in \left[\frac{1}{2}, 1\right]$. The mechanisms build upon the so-called (uniform) permutation mechanism [7, 17], which does not use any predictions and is $\frac{1}{2}$-robust. In a nutshell, this mechanism permutes the vertices uniformly at random and carefully selects a vertex with maximum indegree from vertices that appear previously in the permutation. Our mechanisms favor permutations where the predicted vertex appears towards the end so that most of its incoming edges are likely observed, with a bias that depends on the confidence parameter. We show that for any $\rho \in \left[\frac{1}{2}, 1\right]$ the resulting mechanism is $\rho$-consistent and $(1 - \rho)$-robust (Proposition 3.1), and that this trade-off between consistency and robustness is best-possible (Theorem 6.1). While this optimal consistency–robustness trade-off can also be achieved by the baseline mechanism that randomizes between the uniform permutation mechanism and the mechanism that selects the predicted vertex, such a mechanism would fail basic fairness notions, as discussed in Section 3.

We then study 1-selection mechanisms for plurality voting, where each vertex has exactly one outgoing edge. In this setting, we establish that the 1-permutation mechanism that puts the predicted vertex at the end of the permutation is 1-consistent and $\frac{1}{2}$-robust (Theorem 3.3). Prior work had established that the uniform permutation mechanism is $\frac{2}{3}$-robust, which also implies $\frac{2}{3}$-consistency [11]. By an appropriate lottery between both mechanisms, we achieve $\left(\frac{2}{3} + \frac{1}{3}\rho\right)$-consistency and $\left(\frac{2}{3} - \frac{1}{6}\rho\right)$-robustness for all $\rho \in [0, 1]$ (Corollary 3.4). We further show in Theorem 6.1 that for any $\alpha$-consistent and $\beta$-robust impartial mechanism, $\beta \leq \frac{3}{4}$ and $\alpha + \beta \leq \frac{3}{2}$.

We next consider 2-selection mechanisms. In this setting, the bidirectional permutation mechanism [7] was shown to achieve the optimal robustness guarantee of $\frac{1}{2}$. We show that, by placing the predicted

vertices at both ends of the permutation, we obtain the best-possible consistency guarantee of $1$ without any sacrifice of robustness (Theorem 4.1). For randomized mechanisms, an appropriate lottery between this mechanism and the randomized permutation mechanism [7] achieves $\left(\frac{2}{3} + \frac{1}{3}\rho\right)$-consistency and $\left(\frac{2}{3} - \frac{1}{6}\rho\right)$-robustness for all $\rho \in [0,1]$; see Proposition 4.2. We further show in Theorem 6.1 that, for any $\alpha$-consistent and $\beta$-robust impartial mechanism, $\beta \leq \frac{3}{4}$ and $\alpha + \beta \leq \frac{3}{2}$.

We finally study randomized $k$-selection mechanisms for an arbitrary number $k \in \mathbb{N}$ and obtain Theorem 5.2, our most challenging result in terms of technical difficulty. Bjelde et al. [7] proposed the $k$-partition mechanism with permutation, which partitions the vertices randomly into $k$ sets and selects one vertex from each set in a similar way to the permutation mechanism, but also accounting for edges from outside the set. We propose the $\rho$-partition mechanism for $\rho \in \left[\frac{1}{2}, 1\right]$, that partitions the vertices randomly into $k$ sets but enforces that each set contains exactly one of the predicted vertices. In each set, the predicted vertex is put at position $\rho$ while all other vertices obtain a position drawn uniformly from the unit interval. We then select one vertex from each set, as the $k$-partition mechanism with permutation. The mechanism achieves higher consistency by avoiding that more than one of the predicted vertices ends up in the same set, but the analysis requires new techniques because the probabilities of two optimal vertices being in the same set are no longer independent.

In the realm of mechanisms with predictions, it is common to also study approximation guarantees as a function of the prediction error, commonly referred to as smoothness. In our context, a natural notion of error of a predicted set of vertices is the difference between the maximum indegree of a set of $k$ vertices and the indegree of the predicted set, normalized by the maximum indegree so it lies in the interval $[0,1]$. Since all our $\alpha$-consistent and $\beta$-robust mechanisms provide an $\alpha$-approximation of the indegree of the predicted set, independently of whether this set is or is not optimal, they immediately yield a smoothness guarantee of $\max\left\{\alpha(1-\eta), \beta\right\}$ for an error $\eta \in [0,1]$.

**Related Work.** Impartiality, as we study it here, was first considered by de Clippel et al. [15] for the division of a divisible resource among members of a set of agents based on divisions proposed by the agents. In the context of selection, it was first studied by Holzman and Moulin [19] and Alon et al. [3]. Holzman and Moulin studied deterministic mechanisms for the special case of *plurality voting*, where each member casts exactly one nomination for another member of the set. They showed that, even in this restricted setting, impartiality is incompatible with the axioms of negative and positive unanimity, where the former requires that a member receiving no nomination is never selected and the latter that a member nominated by all members except themselves is always selected. Alon et al. studied the more general setting of approval voting, where members may nominate an arbitrary number of other members and a fixed number $k$ of members is to be selected. Call a mechanism an *exact $k$-selection mechanism* if it always selects exactly $k$ members, and $\alpha$-optimal for $\alpha \in [0,1]$ if the (expected) number of nominations that the selected members receive is always at least an $\alpha$-fraction of the total number of nominations that the $k$ best members receive. In this terminology, Alon et al. showed that no deterministic, impartial, and exact $k$-selection mechanism can be $\alpha$-optimal for any fixed $\alpha > 0$. They further provided a randomized impartial $\frac{1}{4}$-optimal 1-selection mechanism, and a randomized impartial $(1 - o(1))$-optimal $k$-selection mechanism for $k \to \infty$. Fischer and Klimm [17] proposed and analyzed the permutation mechanism and showed that it is $\frac{1}{2}$-optimal, which is best-possible for 1-selection. They further showed that for plurality votes, the same mechanism is $\alpha$-optimal for $\alpha = \frac{67}{108} \approx 0.620$. Cembrano et al. [11] gave a tight analysis of the permutation mechanism for plurality votes, showing that it is even $\frac{2}{3}$-optimal. They further proposed a new mechanism that is $\frac{2105}{3147}$-optimal, where $\frac{2105}{3147} \approx 0.669$. Bjelde et al. [7] showed that deterministic impartial $k$-selection mechanisms that are allowed to sometimes select fewer than $k$ members can perform better than exact $k$-selection mechanisms, and bounded the approximation guarantees of randomized mechanisms that select $k > 1$ members. Caragiannis et al. [9] studied the additive approximation guarantees of impartial selection mechanisms, and Cembrano et al. [12] gave a deterministic mechanism with an improved additive guarantee for plurality votes. Caragiannis et al. [10] considered the additive approximation guarantees of impartial mechanisms that receive *prior information* as additional input. They looked at two different models where members choose their nominations based on a known probability distribution or based on the popularity of a member. We note that this approach differs from ours, since it does not bound the approximation guarantees if the prior information is inaccurate.

The robustness–consistency framework was first used by Purohit et al. [25] to study the performance of online algorithms with predictions. Predictions have been recently incorporated by Berger et al. [6] into the voting setting of metric distortion, where a candidate is to be selected based on rankings

cast by voters with costs given by distances on a common metric space, and the goal is to minimize the ratio between the social cost of the selected candidate and that of the optimal one. More broadly, mechanisms with predictions were first studied by Agrawal et al. [2] for facility location, which has been further considered by Balkanski et al. [5] for randomized mechanisms and different types of predictions, by Fang et al. [16] for a restricted set of candidate locations, and by Istrate and Bonchis [21] for the case where agents' objective is to maximize rather than minimize their distance to the facilities. Balkanski et al. [4] incorporated predictions into the design of strategyproof mechanisms for makespan minimization in scheduling. Xu and Lu [26] also studied a range of mechanism design problems with and without money, including facility location, scheduling, and auction design.

## 2 Preliminaries

For $n \in \mathbb{N}$, let $[n] = \{1, \ldots, n\}$ and let

$$\mathcal{G}_n = \left\{ ([n], E) : E \subseteq ([n] \times [n]) \setminus \bigcup_{i \in [n]} \{(i, i)\} \right\}$$

denote the set of simple graphs with vertex set $[n]$ and without self-loops. For $S, T \in 2^{[n]}$, we denote the edges from vertices in $S$ to vertices in $T$ by

$$N_S^-(T, G) = \{(j, i) \in E : G = ([n], E), j \in S, i \in T\},$$

and the number of such edges by $\delta_S^-(T, G)$. We omit $S$ from the previous notation when $S = [n]$, and we write $N^-(i, G)$ instead of $N^-(\{i\}, G)$ and $\delta^-(i, G)$ instead of $\delta^-(\{i\}, G)$. For $k \in [n]$, we write $\Delta_k(G) = \max_{T \subseteq [n] : |T| = k} \delta^-(T, G)$. We omit $k$ when it is equal to 1 and $G$ whenever it is clear from the context. We refer to the graphs $G = ([n], E) \in \mathcal{G}_n$ such that $|\{(i, j) \in E : j \in [n]\}| = 1$ for every $i \in [n]$, in which all vertices have outdegree exactly one, as *plurality graphs*.

We consider selection mechanisms that obtain a prediction for the set of vertices with maximum indegrees. A *$k$-selection mechanism with predictions* is a family of functions $f \colon \binom{[n]}{k} \times \mathcal{G}_n \to [0, 1]^n$ with $\sum_{i \in [n]} f_i(\hat{S}, G) \leq k$ for all $G \in \mathcal{G}_n$, where $f_i(\hat{S}, G)$ denotes the probability assigned by the mechanism to agent $i$.[2] For a graph $G \in \mathcal{G}_n$ and $i \in [n]$, the number $f_i(\hat{S}, G)$ is the probability that $f$ selects vertex $i$ when $(\hat{S}, G)$ is the input. A mechanism is called *deterministic* if it only assigns probabilities 0 and 1, and is called *impartial* if $f_i(\hat{S}, G) = f_i(\hat{S}, G')$ whenever for two graphs $G = ([n], E)$ and $G' = ([n], E')$ we have $E \setminus \bigcup_{j \in [n]} \{(i, j)\} = E' \setminus \bigcup_{j \in [n]} \{(i, j)\}$.

For $\alpha \in [0, 1]$, we call a $k$-selection mechanism with predictions $\alpha$-consistent if it achieves an $\alpha$-approximation when the predictions are accurate, i.e., $\sum_{i \in [n]} f_i(\hat{S}, G) \delta^-(i, G) \geq \alpha \Delta_k(G)$ for all $n \in \mathbb{N}$, $G \in \mathcal{G}_n$, and $\hat{S} \in \binom{[n]}{k}$ with $\delta^-(\hat{S}, G) = \Delta_k(G)$. For $\beta \in [0, 1]$, we call a $k$-selection mechanism with predictions $\beta$-robust if it achieves a $\beta$-approximation regardless of the predictions' quality, i.e., $\sum_{i \in [n]} f_i(\hat{S}, G) \delta^-(i, G) \geq \beta \Delta_k(G)$ for all $n \in \mathbb{N}$, $G \in \mathcal{G}_n$, and $\hat{S} \in \binom{[n]}{k}$.

We finally require some notation regarding permutations. For a (vertex) set $S$, we let $\Pi_S \subset S^{|S|}$ denote the set of permutations of the set $S$; we refer to the order induced by a permutation as an order *from left to right* for ease of notation. We write $\Pi_n$ as a shorthand for $\Pi_{[n]}$. For a permutation $\pi \in \Pi_S$, a set $S' \subseteq S$, and a vertex $i \in S$, we write $\pi_{<i} = \{j \in S : j = \pi_r, i = \pi_t \text{ for some } r < t\}$ for the set of vertices that appear to the left of $i$, $\pi(S') \in \Pi_{S'}$ for the restriction of $\pi$ to $S'$, and $\bar{\pi} \in \Pi_S$ for the reverse of $\pi$. Sometimes we fix the position of some vertices in the permutation. For a set of distinct vertices $\{i_j : j \in [m]\}$ and distinct positions $\{r_j : j \in [m]\}$, we write $\Pi_S(i_1 \to r_1, \ldots, i_m \to r_m)$ for the set of permutations $\pi \in \Pi_S$ such that $i_j = \pi_{r_j}$ for every $j \in [m]$.

## 3 Selecting a Single Vertex

In this section, we study 1-selection mechanisms with predictions. For ease of notation, we denote the predicted set by $\hat{S} = \{\hat{i}\}$ and write $\Delta(G)$ instead of $\Delta_1(G)$ for the maximum indegree.

It is well known that deterministic mechanisms cannot achieve any constant approximation in the classic setting without predictions, which for our setting has the direct implication that no deterministic

---

[2]It is not hard to see that such a distribution over vertices can be translated into a probability over sets of size at most $k$ via the Birkhoff-von Neumann Theorem; see Bjelde et al. [7, Lemma 2.1] for the details.

| **Algorithm 1** Permutation mechanism $\mathrm{Pm}(G, S, x)$ | **Algorithm 2** $\rho$-permutation mechanism $\mathrm{Pm}^\rho(\hat{\imath}, G)$ |
|---|---|
| **Input:** graph $G = ([n], E)$, set $S \subseteq [n]$, $x \in [0,1]^S$. | **Input:** graph $G = ([n], E)$, predicted vertex $\hat{\imath} \in [n]$. |
| **Output:** vertex $i^{\mathrm{Pm}} \in [n]$. | **Output:** vertex $i^{\mathrm{Pm}} \in [n]$. |

**Algorithm 1**

$\pi \leftarrow \pi(x) \in \Pi_S$
initialize $i^{\mathrm{Pm}} \leftarrow \pi_1$ and $d \leftarrow \delta^-_{[n]\setminus S}(\pi_1)$
**for** $r \in \{2, \dots, |S|\}$ **do**
$\quad i \leftarrow \pi_r$
$\quad$ **if** $\delta^-_{([n]\setminus S)\cup(\pi_{<i}\setminus\{i^{\mathrm{Pm}}\})}(i) \geq d$ **then**
$\quad\quad$ update $i^{\mathrm{Pm}} \leftarrow i$ and $d \leftarrow \delta^-_{([n]\setminus S)\cup\pi_{<i}}(i)$
**return** $i^{\mathrm{Pm}}$

**Algorithm 2**

$x_{\hat{\imath}} \leftarrow \rho$
sample $x_i \in [0,1]$ uniformly at random $\ \forall i \in [n]\setminus\{\hat{\imath}\}$
**return** $\mathrm{Pm}(G, [n], x)$

mechanism with predictions can be $\beta$-robust for a constant $\beta > 0$. Thus, the trivial answer to the best-possible trade-off between consistency and robustness is given by the mechanism that selects the predicted vertex $\hat{\imath}$ and achieves 1-consistency and 0-robustness.

The problem becomes more interesting with randomization, as the best-known mechanism for the setting without predictions achieves a $\frac{1}{2}$-approximation. We refer to the mechanism achieving this approximation, introduced by Fischer and Klimm [17], as the *uniform permutation mechanism*. This mechanism sorts the vertices uniformly at random and considers them one by one according to this order while maintaining a candidate vertex, initially the first vertex. A vertex is taken as the new candidate if its observed indegree is larger than that of the current candidate, where *observed indegree* refers to the indegree when only considering incoming edges from previous vertices and omitting a potential edge from the current candidate. The vertex that is the candidate in the end is selected.

We define a more general version of this mechanism, where in addition to the graph $G = ([n], E)$, the mechanism receives a subset of vertices $S \subseteq [n]$ and a vector $x \in [0,1]^S$. Vertices in $S$ are those taken into account for the permutation, while all other vertices in $[n] \setminus S$ are not eligible for selection and the incoming edges from these vertices are always considered. The vector $x \in [0,1]^S$ defines the permutation $\pi \in \Pi_S$: $i$ comes before $j$ if its associated value $x_i$ is smaller than $x_j$. Formally, for every $i, j \in S$ we have $i \in \pi_{<j}$ if and only if either $x_i < x_j$ or both $x_i = x_j$ and $i < j$ hold (we break ties in favor of vertices with smaller indices). We denote the permutation $\pi \in \Pi_S$ constructed in this way from $x \in [0,1]^S$ by $\pi(x)$.

The permutation mechanism for a fixed set $S$ and vector $x \in [0,1]^S$ is formally described in Algorithm 1; we refer to its output for a graph $G$, a set $S$, and a vector $x$ by $\mathrm{Pm}(G, S, x)$. The uniform permutation mechanism, providing the best-possible guarantee among randomized 1-selection mechanisms without prediction, corresponds to the mechanism that receives a graph $G$ and returns $\mathrm{Pm}(G, [n], x)$, where $x_i \in [0,1]$ is taken uniformly at random for each $i \in [n]$.

Instead of the uniform permutation mechanism, we consider in the setting with predictions the $\rho$-permutation mechanism, given in Algorithm 2. This mechanism receives a graph $G = ([n], E)$ and a predicted vertex $\hat{\imath} \in [n]$, and returns $\mathrm{Pm}(G, [n], x)$, where now $\hat{\imath}$ has an associated value $x_{\hat{\imath}} = \rho$ and all values $x_i$ for $i \in [n] \setminus \{\hat{\imath}\}$ are sampled uniformly at random. The value $\rho$ then has the natural interpretation of a confidence parameter: Taking $\rho = 1$ ensures seeing all incoming edges of the predicted vertex, while smaller values of $\rho$ increase the probability of seeing potential outgoing edges of $\hat{\imath}$. This mechanism attains any convex combination of $\alpha$-consistency and $\beta$-robustness between the points $(\alpha, \beta) \in \left\{(1, 0), (\frac{1}{2}, \frac{1}{2})\right\}$. In Section 6, we will see that this trade-off is actually best-possible.

**Proposition 3.1.** *For any confidence parameter $\rho \in \left[\frac{1}{2}, 1\right]$ the $\rho$-permutation mechanism is impartial, $\rho$-consistent and $(1 - \rho)$-robust.*

We need some notation. For a fixed graph $G = ([n], E) \in \mathcal{G}_n$, set $S \subseteq [n]$, and vector $x \in [0,1]^S$, we let $i^{\mathrm{Pm}}(G, S, x)$ denote the outcome of $\mathrm{Pm}(G, S, x)$. Whenever $x$ is fixed, we write $\pi$ for the induced permutation instead of $\pi(x)$. As a key property for the analysis of the (uniform) permutation mechanism, Bousquet et al. [8] showed that, for any fixed permutation, it selects a vertex with maximum indegree from the left. Bjelde et al. [7] extended this result to the case where we restrict to a set of vertices and consider all incoming edges from other vertices. We phrase the latter result with our notation as the following lemma, which we apply in the full version to prove Proposition 3.1.

**Lemma 3.2** (Bjelde et al. [7]). *For every $G = ([n], E) \in \mathcal{G}_n$, $S \subseteq [n]$, and $x \in [0,1]^S$, it holds that $i^{\mathrm{Pm}}(G, S, x) \in \arg\max\{\delta_{([n]\setminus S)\cup\pi_{<i}}(i, G) : i \in [n]\}$.*

It is worth noting that the consistency and robustness guarantees of Proposition 3.1 are also achieved by a baseline mechanism that returns the predicted vertex with probability $\rho$ and runs the uniform permutation mechanism with probability $1 - \rho$. However, the baseline mechanism fails a basic *unanimity* notion introduced by Holzman and Moulin [19]: If a vertex $v$ is such that all other vertices have a single outgoing edge to $v$, then $v$ should be selected. Whenever $v$ is not the predicted vertex, the baseline mechanism fails to select $v$ with constant probability, while the $\rho$-permutation mechanism returns $v$ as long as it is not first or second in the permutation, i.e., with probability $1 - O\left(\frac{1}{n}\right)$.

**Plurality Voting.** A usual restriction in voting is that each member nominates one other member, which in our graph representation implies having vertices with outdegree one. This paradigm of *plurality voting*, extensively considered in the impartial selection literature [11, 19, 23], has been shown to enable better approximation guarantees for randomized mechanisms.[3] In particular, Cembrano et al. [11] proved that the uniform permutation mechanism provides an improved approximation ratio of $\frac{2}{3}$ in this case.

In our setting, we show that the $\rho$-permutation mechanism with $\rho = 1$, where the predicted vertex is deterministically placed at the end of the permutation and all other vertices are sorted uniformly at random, achieves 1-consistency and $\frac{1}{2}$-robustness. The following theorem provides a more fine-grained bound on the robustness of this mechanism as a function of the maximum indegree $\Delta$ of the input graph; the bound of $\frac{1}{2}$ follows by taking the worst case over $\Delta$.

**Theorem 3.3.** *The 1-permutation mechanism is impartial, 1-consistent, and $\beta(\Delta)$-robust on plurality graphs with maximum indegree $\Delta \geq 2$, where*

$$\beta(\Delta) = \begin{cases} \frac{3\Delta - 2}{4\Delta} & \text{if } \Delta \text{ is even,} \\ \frac{3\Delta^2 - 2\Delta - 1}{4\Delta^2} & \text{if } \Delta \text{ is odd.} \end{cases}$$

*Moreover, this function $\beta$ is increasing, implying that this mechanism is impartial, 1-consistent, and $\frac{1}{2}$-robust on plurality graphs.*

We prove this theorem in the full version using a strengthened version of a lemma of Cembrano et al. [11] that establishes a negative correlation between the indegree from the left of the maximum-indegree vertex and that of all other vertices. We show that this result remains true for the non-uniform distribution over permutations induced by the vector $x$ defined in the 1-permutation mechanism, and that it holds not only for the maximum-indegree vertex but for any fixed vertex. The proof adapts that of Cembrano et al. [11], defining an injective function between sets of permutations to couple the probabilities that certain indegrees are observed in the permutation taken in the mechanism. We then use the lemma to prove Theorem 3.3. The most challenging case, which ultimately leads to a worse robustness guarantee than in the setting without predictions, is when the maximum-indegree vertex has an incoming edge from the predicted vertex, as this edge is never considered by the mechanism when observing the indegrees from the left. However, since all outdegrees are 1, we can still obtain a lower bound on the probability of selecting this maximum-indegree vertex or another vertex with high indegree.

We now state the implications of Theorem 3.3, in terms of the trade-off between consistency and robustness we can achieve by combining the 1-permutation mechanism with the uniform permutation mechanism. The proof of this result can be found in the full version. In Section 6, we will see that this trade-off is not far from tight.

**Corollary 3.4.** *For every $\rho \in [0, 1]$, there exists a randomized 1-selection mechanism with predictions that is impartial, $\alpha$-consistent, and $\beta$-robust on plurality graphs with maximum indegree $\Delta$, where*

$$\alpha(\Delta) = \begin{cases} \frac{3\Delta+2}{4(\Delta+1)} + \frac{\Delta+2}{4(\Delta+1)}\rho & \text{if } \Delta \text{ is even,} \\ \alpha(\Delta - 1) & \text{if } \Delta \text{ is odd,} \end{cases} \qquad \beta(\Delta) = \begin{cases} \frac{3\Delta+2}{4(\Delta+1)} - \frac{\Delta+2}{4\Delta(\Delta+1)}\rho & \text{if } \Delta \text{ is even,} \\ \frac{3\Delta-1}{4\Delta} - \frac{\Delta+1}{4\Delta^2}\rho & \text{if } \Delta \text{ is odd.} \end{cases}$$

*In particular, for every $\rho \in [0, 1]$, there exists a randomized 1-selection mechanism with predictions that is impartial, $\left(\frac{2}{3} + \frac{1}{3}\rho\right)$-consistent, and $\left(\frac{2}{3} - \frac{1}{6}\rho\right)$-robust on plurality graphs.*

---

[3]The impossibility of providing a constant approximation of the maximum indegree with deterministic mechanisms remains true in this restricted setting [19].

---

**Algorithm 3** Fixed bidirectional permutation mechanism, $\text{Pm}_{\text{bi}}(\hat{S}, G)$

---

**Input:** graph $G = ([n], E)$, predicted set $\hat{S} = \{\hat{\imath}_1, \hat{\imath}_2\} \subseteq [n]$.
**Output:** set $S \subseteq [n]$ with $|S| \leq 2$.
    Fix $x_{\hat{\imath}_1} \leftarrow 0$ and $x_{\hat{\imath}_2} \leftarrow 1$
    fix $x_i \in (0, 1)$ arbitrarily for each $i \in [n] \setminus \hat{S}$
    $\bar{x}_i \leftarrow 1 - x_i$ for every $i \in [n]$
    **return** $\text{Pm}(G, [n], x) \cup \text{Pm}(G, [n], \bar{x})$

---

## 4 Selecting Two Vertices

In this brief section, we state our results for the selection of two vertices. In terms of mechanisms without predictions, the best-known deterministic and randomized impartial mechanisms achieve $\frac{1}{2}$- and $\frac{2}{3}$-optimality, respectively. While the bound for deterministic mechanisms is best-possible, only an upper bound of $\frac{3}{4}$ is known for randomized mechanisms [7]. For compactness, throughout this section we denote the predicted set by $\hat{S} = \{\hat{\imath}_1, \hat{\imath}_2\}$.

The deterministic mechanism achieving $\frac{1}{2}$-optimality is based on the permutation mechanism. It runs, for an arbitrarily fixed permutation $\pi$, the permutation mechanism for both $\pi$ and its reverse $\bar{\pi}$, and returns the selected vertices for each direction (potentially the same vertex). A natural approach to incorporate the prediction is to run this mechanism with the predicted vertices at both extremes of the fixed permutation. The resulting mechanism, which we call *fixed bidirectional permutation*, maintains the best-possible robustness of $\frac{1}{2}$ while achieving 1-consistency. The formal description of the mechanism is given in Algorithm 3; the proof of this result is deferred to the full version.

**Theorem 4.1.** *The fixed bidirectional permutation mechanism is impartial, 1-consistent, and $\frac{1}{2}$-robust.*

In terms of randomized mechanisms, convex combinations of the best-known mechanism without prediction, achieving $\frac{2}{3}$-robustness [7], and the fixed bidirectional permutation mechanism, achieving 1-consistency and $\frac{1}{2}$-robustness, allows us to attain combinations of $\alpha$-consistency and $\beta$-robustness between $(\alpha, \beta) = \left(\frac{2}{3}, \frac{2}{3}\right)$ and $(\alpha, \beta) = \left(1, \frac{1}{2}\right)$. We state this simple fact in the following proposition; we will see in Section 6 that this combination of consistency and robustness is not far from tight.

**Proposition 4.2.** *For every $\rho \in [0, 1]$, there exists a randomized 2-selection mechanism with predictions that is impartial, $\left(\frac{2}{3} + \frac{1}{3}\rho\right)$-consistent, and $\left(\frac{2}{3} - \frac{1}{6}\rho\right)$-robust.*

## 5 Selecting $k \geq 3$ Vertices

In this section, we study the impartial selection of $k \geq 3$ vertices when the mechanism is equipped with a prediction on the optimal set.

In terms of deterministic mechanisms, the setting without predictions is far from well understood. Indeed, a large gap remains between the best-known lower and upper bounds of $\frac{1}{k}$ and $\frac{k-1}{k}$ on the approximation guarantee that impartial mechanisms can achieve [7]. Recently, Cembrano et al. [13] improved the lower bound for cases where $k$ is larger than (approximately) $2\sqrt{n}$, but the lower bound of $\frac{1}{k}$ remains the best-known bound for an arbitrary number of agents $n$. This guarantee comes from the bidirectional permutation mechanism explained in the previous section, whose $\frac{1}{2}$-approximation of the optimal set of two agents translates into a $\frac{1}{k}$-approximation of the optimal committee of $k$ agents. Similarly to the previous section, we can modify this mechanism to maintain its robustness guarantee and achieve 1-consistency. Specifically, we select $k - 2$ vertices from the predicted set and one or two more vertices through our fixed bidirectional permutation mechanism, with the remaining two predicted vertices at the extremes of the permutation. We state the properties of this simple mechanism in the following proposition, proven in the full version.

**Proposition 5.1.** *There exists a deterministic $k$-selection mechanism with predictions that is impartial, 1-consistent, and $\frac{1}{k}$-robust.*

Regarding randomized mechanisms, the best-known mechanism for $k$-selection was developed by Bjelde et al. [7] and provides an approximation guarantee of $\frac{k}{k+1}\left(1 - \left(\frac{k-1}{k}\right)^{k+1}\right)$, which starts at

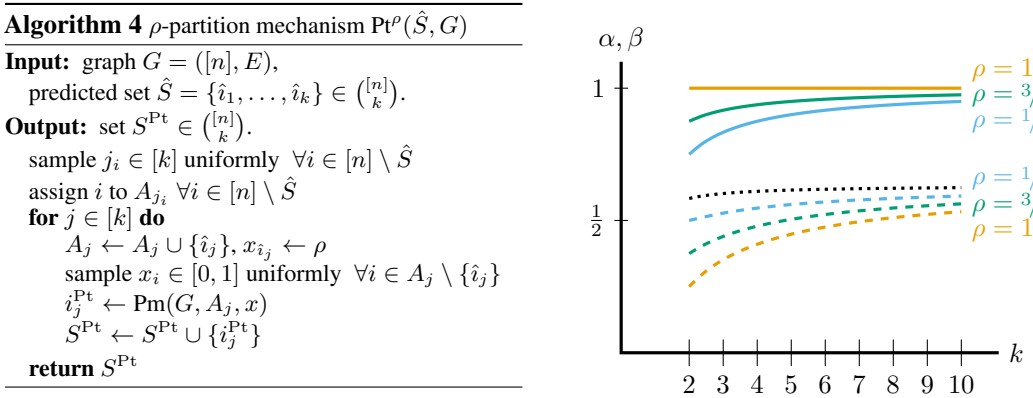

**Algorithm 4** $\rho$-partition mechanism $\text{Pt}^\rho(\hat{S}, G)$

---

**Input:** graph $G = ([n], E)$,
  predicted set $\hat{S} = \{\hat{\imath}_1, \ldots, \hat{\imath}_k\} \in \binom{[n]}{k}$.
**Output:** set $S^{\text{Pt}} \in \binom{[n]}{k}$.
  sample $j_i \in [k]$ uniformly $\forall i \in [n] \setminus \hat{S}$
  assign $i$ to $A_{j_i}$ $\forall i \in [n] \setminus \hat{S}$
  **for** $j \in [k]$ **do**
    $A_j \leftarrow A_j \cup \{\hat{\imath}_j\}, x_{\hat{\imath}_j} \leftarrow \rho$
    sample $x_i \in [0,1]$ uniformly $\forall i \in A_j \setminus \{\hat{\imath}_j\}$
    $i_j^{\text{Pt}} \leftarrow \text{Pm}(G, A_j, x)$
    $S^{\text{Pt}} \leftarrow S^{\text{Pt}} \cup \{i_j^{\text{Pt}}\}$
  **return** $S^{\text{Pt}}$

---

Figure 2: The $\rho$-partition mechanism (left) and a plot of its $\alpha$-consistency (solid) and $\beta$-robustness (dashed) for the values $\rho = 1$, $\rho = \frac{3}{4}$, and $\rho = \frac{1}{2}$ as a function of $k$ (right). The dotted black line is the consistency and robustness of the $k$-partition mechanism of Bjelde et al. [7].

$\frac{7}{12} \approx 0.5833$ for $k = 2$, $\frac{65}{108} \approx 0.6019$ for $k = 3$, and approaches $1 - \frac{1}{e} \approx 0.6321$ as $k$ grows. The mechanism assigns each vertex to one out of $k$ sets uniformly at random. It then selects one vertex from each set via the permutation mechanism restricted to that set with an internal permutation taken uniformly at random. While its impartiality is easy to see, the approximation guarantee requires a careful analysis of the expected observed indegree of optimal vertices in each set. In the following, we develop a randomized mechanism with predictions inspired by this mechanism that achieves almost optimal robustness while losing very little in terms of consistency, especially as $k$ grows.

As in the mechanism by Bjelde et al., vertices are assigned to one of $k$ sets, and one vertex is selected from each set by running the permutation mechanism restricted to the set. However, both the assignment to sets and the permutation are not taken independently and uniformly for each vertex anymore. Instead, we assign one predicted vertex to each set; all other vertices are still assigned to a set chosen independently and uniformly at random. Within each set, the permutation is sampled as in the $\rho$-permutation mechanism from Section 3: For each set $A_j$ with a predicted vertex $\hat{\imath}_j$, we take a vector $x \in [0,1]^{A_j}$ such that $x_{\hat{\imath}_j} = \rho$ and $x_i \in [0,1]$ is taken uniformly at random for each $i \in A_j \setminus \{\hat{\imath}_j\}$. Intuitively, these changes allow the mechanism to see most incoming edges of the predicted vertices while only mildly affecting the distributions to keep a strong robustness guarantee.

The mechanism, which we refer to as the $\rho$-partition mechanism, is formally presented in Algorithm 4. For $\rho \in [0,1]$, we denote its output by $\text{Pt}^\rho(\hat{S}, G)$ for each graph $G = ([n], E)$ and predicted set $\hat{S}$. By tuning the confidence parameter $\rho$ between $\frac{1}{2}$ and 1, we achieve a consistency between $1 - \frac{1}{2k}$ and 1 while only losing $O\left(\frac{1}{k}\right)$ in robustness compared to the best-known mechanism without prediction.

**Theorem 5.2.** *For any confidence parameter* $\rho \in \left[\frac{1}{2}, 1\right]$, *the* $\rho$-*partition mechanism is impartial,* $\alpha$-*consistent, and* $\beta$-*robust, where* $\alpha = 1 - \frac{1-\rho}{k}$ *and* $\beta = \left(1 - \frac{2\rho}{k+1}\right)\left(1 - \left(\frac{k-1}{k}\right)^k\right)$.

For example, when taking $\rho = \frac{1}{2}$ to prioritize robustness, our mechanism achieves a robustness guarantee of $\frac{1}{2}$ for $k = 2$, $\frac{19}{36} \approx 0.5278$ for $k = 3$, $\frac{35}{64} \approx 0.5469$ for $k = 4$, and approaching $1 - \frac{1}{e} \approx 0.6321$ for $k \to \infty$. The consistency guarantee for this value of $\rho$ and any $k \geq 2$ is $1 - \frac{1}{2k}$, which is $\frac{3}{4} = 0.75$ for $k = 2$, $\frac{5}{6} \approx 0.8333$ for $k = 3$, $\frac{7}{8} = 0.875$ for $k = 4$, and approaches 1 for $k \to \infty$. When taking $\rho = 1$ to maximize consistency, the mechanism is 1-consistent for any $k$ and achieves a robustness guarantee of $\frac{1}{4} = 0.25$ for $k = 2$, $\frac{19}{54} \approx 0.3519$ for $k = 3$, $\frac{105}{256} \approx 0.04102$ for $k = 4$, and again approaching $1 - \frac{1}{e} \approx 0.6321$ for $k \to \infty$. Figure 2 illustrates the performance of the $\rho$-partition mechanism for $\rho \in \left\{\frac{1}{2}, \frac{3}{4}, 1\right\}$ and $k \in \{2, \ldots, 10\}$, and compares it with the $k$-partition mechanism of Bjelde et al. [7].

The proof of Theorem 5.2 can be found in the full version; here we briefly describe the main ideas behind the robustness guarantee, which constitutes the most difficult part of the proof. For the analysis we consider an optimal set $S^*$ and $j \in [k]$ such that $A_j$ contains an optimal vertex, i.e., $S^* \cap A_j \neq \emptyset$, and sample a vertex $i^*$ from $S^* \cap A_j$ uniformly at random. We then bound the expected indegree of $i^*$ that the mechanism observes by bounding the probability that each in-neighbor $i$ of $i^*$ lies in a set other than $A_j$ or in the set $A_j$ but before $i^*$ according to the internal permutation. What complicates

the analysis is that, unlike in the mechanism without predictions, the events $i^* \in A_j$ and $i \in A_j$ are not independent. However, it is not difficult to see that when $i \notin S^* \cup \hat{S}$, the probability of $i$ being in $A_j$ is the same as in the independent case. We show further that when $i \in S^*$ or $i^* \in \hat{S}$, the probability of $i$ being in $A_j$ cannot increase much, and the only difference is given by the position of the predicted vertex in the internal permutation. The most intricate part of the proof is the case where $i^* \in S^* \setminus \hat{S}$ and $i \in \hat{S} \setminus S^*$, because the events of $i^*$ being sampled in the set $A_j$ and $i$ being in this set can be strongly correlated. Indeed, the probability of the former event conditional on $i \in A_j$ can be as large as 1 if, for example, all predicted vertices except $i$ belong to $S^*$, as in this case $i \in A_j$ implies that $i^*$ is the unique vertex in $S^* \cap A_j$. We tackle this difficulty by directly computing a lower bound on the (unconditional) probability of $i^*$ being sampled as the optimal vertex in $A_j$.

## 6 Upper Bounds

To put our consistency and robustness results into perspective, we will now give upper bounds on the values $\alpha$ and $\beta$ for which an impartial selection mechanism with predictions can simultaneously guarantee $\alpha$-consistency and $\beta$-robustness. We do so for $k$-selection with $k \in \{1, 2, 3\}$, and for 1-selection from plurality graphs. The upper bounds are shown in Figure 1 alongside the lower bounds obtained in earlier sections.

**Theorem 6.1.** *The following statements hold:*

(i) *If a randomized 1-selection mechanism with predictions is impartial, $\alpha$-consistent, and $\beta$-robust, then $\beta \leq \frac{1}{2}$ and $\alpha + \beta \leq 1$.*

(ii) *If a randomized 1-selection mechanism with predictions is impartial, $\alpha$-consistent, and $\beta$-robust on plurality graphs, then $\beta \leq \frac{3}{4}$ and $\alpha + \beta \leq \frac{3}{2}$.*

(iii) *If a randomized 2-selection mechanism with predictions is impartial, $\alpha$-consistent, and $\beta$-robust, then $\beta \leq \frac{3}{4}$ and $\alpha + \beta \leq \frac{3}{2}$.*

(iv) *If a randomized 3-selection mechanism with predictions is impartial, $\alpha$-consistent, and $\beta$-robust, then $\beta \leq \frac{4}{5}$, $4\alpha + 3\beta \leq 6$, and $4\alpha + 21\beta \leq 20$.*

We prove these results in the full version. To this end, we consider appropriate families of graphs and for each vertex in these graphs introduce a variable for the probability with which some impartial, $\alpha$-consistent, and $\beta$-robust $k$-selection mechanism selects that vertex. We generalize a lemma of Holzman and Moulin [19] to show that one can restrict attention to symmetric mechanisms, and use impartiality, consistency, robustness, and the fact that the probabilities for each graph must sum up to $k$ to obtain a set of linear inequalities involving the probability variables, $\alpha$, and $\beta$. We then show that any values of $\alpha$ and $\beta$ *not* satisfying the statements violate the linear inequalities.

## 7 Discussion

We have initiated the study of impartial selection mechanisms with predictions. Unlike majority voting, these mechanisms are not prone to strategic manipulation. While we have made substantial progress regarding the approximation guarantees achievable by such mechanisms, in most settings a moderate gap remains between the upper and lower bounds. We leave closing these gaps for future work. In addition, it would be interesting to test the mechanisms we have proposed in practical applications, for example in the aggregation of outputs of different LLMs.

## Acknowledgements

Research was supported by the Deutsche Forschungsgemeinschaft under project number 431465007, by the Engineering and Physical Sciences Research Council under grant EP/T015187/1, and by a Structural Democracy Fellowship through the Brooks School of Public Policy at Cornell University.

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
