# OpenReview forum: "Impartial Selection with Predictions"
_NeurIPS.cc/2025/Conference — NeurIPS 2025 poster_

### Official Review · Reviewer_mkRF · 2025-06-27

**Clarity:** 2
**Significance:** 3
**Originality:** 2
**Rating:** 4
**Confidence:** 3

**Summary:**

Impartial Selection mechanisms are voting mechanisms that are largely used to select agents based on nominations provided by the agents themselves. They have been very-well studied in the community of Social Choice and are used in several real contexts, such as committee selection.
This paper starts the study of such voting mechanisms in a setting where the mechanism receives in input  a “suggestion” of an optimal solution but the precision of the suggestion is not guaranteed. In the last years the area of mechanism design with prediction received a lot of interests and this framework has been applied to a long list of important problems. The increasing interest on this framework is due to the possibility of using it to model data-driven mechanisms, where the mechanism may receive suggestions from an expert or from some LLMs.

The paper focuses on the question of whether impartial selection mechanisms can be improved if they are given in input a prediction of a set of agents receiving a high number of nominations. Extensions of known mechanisms to the setting of mechanisms with prediction are presented both for the general setting, where each agent can present an arbitrary number of nominations and k agents have to be elected, and for more restricted settings. For all these settings bounds are given on the consistency and robustness.

**Questions:**

1.	What is the meaning of the superscript used in your notation? (see for example lines 171-172). In general, I think your notation is not intuitive and too heavy.
2.	At line 178, which is the definition of the function f and its relation to the f_i? Since f_i is a conditional probability, why don’t use the traditional notation for probabilities?
3.	At line 216, I suggest to directly denote the permutation as \pi(x), as done at line 219. Set S is implicitly defined by the vector x and you can remove it from the inputs to the mechanism.
4.	At lines 229-231: your assumption is not sufficient to guarantee that the predicted vertex i will be the last vertex in the permutation and the mechanism can see all its incoming. For example, assume that x_j = 1 and i<j. Why is the statement at line 231 true? (Similar observations hold for the selection mechanisms proposed in the following sections for k>1).
5.	In section 6 you give bounds on consistency and robustness achievable by impartial mechanisms only for k<=3. Any idea on what happens for k>3? Why is your argument not working?

**Ethical Concerns:**

["NO or VERY MINOR ethics concerns only"]

**Final Justification:**

Authors answered all my questions in a satisfactorily way. I confirm my positive feeling on the paper.

**Limitations:**

yes.

**Paper Formatting Concerns:**

No major formatting issue observed.

**Quality:**

2

**Strengths And Weaknesses:**

STRENGHTS
The paper studies an important (known) problem in the social choice literature, with significant practical applications, in the setting of mechanisms with prediction. The argument perfectly fits the scopus of the conference.
This is a solid theoretical paper, with interesting contributions that adapt previously known mechanisms to the new setting of mechanisms with prediction.
The quality of writing is pretty good.

WEAKNESSES
The contribution of the paper is exclusively theoretical and mainly incremental. It takes known mechanisms for impartial selection without prediction and adapts them to the setting of mechanisms with prediction.

The extensions are quite trivial but the analysis is not so simple and authors had to solve some technical challenges. Technical proofs are completely moved to the appendix and very few elements are given in the paper to convince the reader of the soundness of the statements. Moreover, appendix is simply a list of proofs of lemmas and theorems with no intuition of how the argument works.

An experimental analysis to validate the practical effectiveness of the proposed mechanisms is totally missing and left as future work.

---

> ### Author Rebuttal · Authors · 2025-07-29
>
> We appreciate your thorough review and detailed comments. We agree that more intuition for the ideas involved in the proofs would be helpful to the reader, and we would be happy to use the extra page allotted for the final version for this purpose.
>
> Regarding the questions:
>
> - The superscript $-$ is used in $N^-$ and $\delta^-$ to denote the in-neighborhood and indegree of a vertex, as standard in the literature to distinguish from out-neighborhood and outdegree. Since we do not need notation for the latter objects, we agree that removing it could make the notation a bit lighter.
>
> - $f_i(\hat{S},G)$ denotes the $i$th entry of $f(\hat{S},G)$, i.e., the probability assigned by a mechanism $f$ to a vertex $i$. We will make this clearer in the next iteration of the manuscript. We have chosen this notation rather than the traditional notation for probabilities for compactness,  since a mechanism is naturally defined as a function (from a pair graph/predicted set to a random subset of vertices) and must fully determine the selection probabilities.
>
> - We are glad to take the suggestion for the notation $\pi(x)$ and the set $S$. Regarding the values $x_j$ for non-predicted vertices, they are sampled uniformly from $[0,1]$ in all randomized mechanisms based on this approach, so $x_j=1$ is a zero-probability event and does not affect the performance guarantee, but we agree this can be more clearly stated. In the (deterministic) fixed bidirectional permutation mechanism, we take $x_j\in (0,1)$ to avoid this case.
>
> - The upper bounds in Section 6 have been obtained by computing mechanisms with optimal robustness and consistency guarantees and constructing graphs showing optimality (this works via LP duality). The same approach can be used to obtain upper bounds for $k>3$, but the difficulty increases and confidence in the tightness of the bounds decreases as $k$ grows. We have given a selection of bounds that we think are most interesting given our current state of knowledge but will make sure to provide a clearer justification for our choices: the case $k=1$ is solved; for $k=2$ we can achieve the best possible robustness subject to $1$-consistency, but cannot currently do the same when $1$-consistency is relaxed; the case of $k=2$ gives rise to the same bounds as the case of plurality for $k=1$, which suggests a connection between the two settings; for $k\geq 3$ the optimal Pareto frontier and the currently achievable Pareto frontier do not intersect, and both are more complex than those for $k=2$.

---

### Official Review · Reviewer_mVCz · 2025-06-30

**Clarity:** 4
**Significance:** 2
**Originality:** 3
**Rating:** 4
**Confidence:** 3

**Summary:**

Consider as input a directed graph with n vertices. The goal is to randomly select k vertices that approximately maximize the sum of in-degrees. The sum of in-degrees corresponds to the utilitarian welfare that the k vertices (agents) provide to the population of n vertices (agents), where an edge from agent i to j means agent i approves of j being selected. Additionally, as part of the input, we are given k proposed vertices S, which are predicted to form a good committee. Now we want to randomize in such a way that:
 (1) if the prediction is good, then the returned outcome is (almost) as good as the prediction (2) is the prediction is bad, then the outcome should not be (much) worse than if we had not seen the prediction and (3) an agent's probability of being selected should be independent of her votes/ her out degree - in other words the mechanism should be impartial.


In this setting, the authors prove the following results:
k=1:
A baseline algorithm for this problem is randomizing between the prediction and the best known algorithm without prediction and achieves \rho-consistency and (1-\rho) robustness. The authors show that this is tight. However, this algorithm has the property that if every node points to the same node v, then still the algorithm would only select v with constant probability, if v is not the predicted vertex. Thus the authors propose an alternative mechanism that selects such a node v w.p. 1-O(1/n) and otherwise has the same consistency and robustness guarantees as the baseline. They call the new mechanism \rho-permutation mechanism.

k=2:
The authors extend the existing deterministic mechanism to the fixed bidirectional permutation mechanism, which is impartial, 1-consistent and 1/2-robust.

k>=:3 The authors introduce a new mechanism called \rho-partition, parametrized by \rho that is impartial and is alpha consistent and beta-robust for \alpha and \beta which for large k tend to 1 and 1-1/e respectively. This algorithm is inspired by the best known k-selection algorithm.

The authors also prove results on how large the alpha and beta can be for an impartial randomized mechanism for k=1,2,3.

**Questions:**

How can a mechanism with predictions that come from black box methods like LLMs that run weird heuristics internally be impartial? Can you model your predictions so that your formal model makes sense and yet is still well-motivated in practice?

Can you explain how your results can help in AI alignment in more detail, also with reference to the less wrong post you link?

**Ethical Concerns:**

["NO or VERY MINOR ethics concerns only"]

**Final Justification:**

This is a good paper and the authors clarified one thing I found confusing.

**Limitations:**

The major limitation is that prediction framework and and impartial mechanism don't naturally fit together.

**Quality:**

3

**Strengths And Weaknesses:**

The paper is very well written and structured. Incorporating prediction in algorithms in light of the arrival of LLMs, which may do better by following strange black-box heuristics, has been a recent hot topic.
The AI-Safety motivation in the introduction is very interesting, albeit the detailed mapping to the problem would need to be elaborated upon.
Moreover, the paper has technically quite non-trivial results and makes good progress in this direction, with the main result being for k ≥ 3.
One weakness is that the baseline algorithm for k = 1 already achieves the desirable optimal trade-off between robustness and consistency.
In general, I think this is a good paper, if not for one conceptual concern regarding the motivation:

Especially in optimization problems, I believe using the prediction framework, given that we are in the age of ML-heuristics, makes a lot of sense. However, I feel that once we add impartiality, the framework no longer makes sense, at least not in the way it is usually motivated. Suppose an AI agent, like an LLM, predicts an outcome k, which will be the input to your algorithm. Since this AI is good at prediction, presumably even without access to the agent’s preferences it will be good at predicting them and might use them to construct its prediction. It is a black-box machine; we simply don’t know. So, for all we know, the prediction algorithm itself might not be impartial (to the extent that the AI is also good at predicting preferences and potentially uses them) thus making the entire mechanism partial. I feel that this is a major flaw in your formal model.

---

> ### Author Rebuttal · Authors · 2025-07-29
>
> Thanks for asking for clarification regarding the impartiality of the prediction. This is an important issue, and we will revise the paper to discuss it in more detail.
> We think that there are many situations where the prediction is clearly impartial since it simply does not have access to the votes of the agents.
> As an example, consider a scenario where a group of agents cast votes on one another about who should be considered for a promotion.
> They can feed the CVs of everybody to an LLM, asking it for its opinion on who is most eligible and taking its output as a prediction. (For the following arguments, it is immaterial whether the agents know the prediction of the LLM or not.)
> In a second step, they can use one of the mechanisms in this paper to do a formal vote.
> This two-step process has the advantage that when the output of the LLM does not align at all with the opinions of the group, they can overrule its decision.
> In addition, the process is impartial in the sense that nobody can influence their own chance of being promoted.
>
> Regarding the use of impartial mechanisms in AI alignment, the Less Wrong post discusses the idea of two AI entities evaluating whether the other one is aligned with human values or not, with the ultimate consequence that an AI entity that is not aligned will be destroyed.
> When slightly rephrasing this thought experiment as one with $n$ AIs that are tasked with evaluating the alignment of the other entities, this is an impartial selection problem where no AI wants to be selected for destruction.
> Using an impartial selection mechanism ensures that the opinion an AI entity voices regarding the alignment of other entities does not influence whether this AI entity is selected for destruction. We think that this is a useful property to incentivize omnipotent AIs to reveal their true evaluation of the other AIs.
> While this scenario does not involve predictions, and we were using it to argue that impartial selection in general is useful for AI alignment, one may alter the process and for example incorporate human expert advice as an external prediction on the rogueness of AIs.

---

> > ### Comment · Reviewer_mVCz · 2025-08-06
> >
> > Thank you for the response! Suppose we have some data D such as the voters CVs and we have the preferences of the voters P. Suppose also the data D is an excellent predictor of the voters true preferences P (maybe there is even a causal relationship).
> > Then the AI/LLM could get the data D, from its model of human values infers the preferences P (perhaps approximately) and then select the social welfare maximizing voter. The prediction would thus not be impartial. Thus, I don't see why in your example, the resulting selection is impartial.

---

> > > ### Author Response · Authors · 2025-08-08
> > >
> > > Thank you!
> > >
> > > As in any mechanism design setting there is a distinction between true preferences (with respect to which we want to optimize) and revealed preferences (revealed to the mechanism by self-interested agents and not necessarily truthful). The crucial point in a setting with predictions is that the prediction is a prediction (hopefully, but not necessarily, accurate) of the true preferences but not influenced by the revealed preferences. Impartiality is the property that no agent can manipulate their revealed preferences (in our setting, their stated opinion of others) to their own benefit (their chance of being selected).
> > >
> > > Here's how this plays out for the 1-permutation mechanism (the argument would be similar for the other mechanisms we discuss). Suppose the AI/LLM parses the CVs and finds out that agent 1 is best qualified for promotion. Then the mechanism would put this agent to the right of a random permutation of the other agents. It then only counts votes from left to right, which ensures that votes are only counted once the voting agent cannot be selected anymore. Even if we communicated to the agents that agent 1 is the rightmost in the permutation, and even if we communicated in addition the random permutation of the other agents, no agent can increase their own chance of being selected by misreporting their opinion. This is regardless of the fact whether agents think that agent 1 or any other agent is eligible for promotion and simply relies on the fact that the votes of an agent are only looked at once the mechanism decided that this agent will not be selected.

---

### Official Review · Reviewer_jF4z · 2025-06-30

**Clarity:** 4
**Significance:** 3
**Originality:** 3
**Rating:** 5
**Confidence:** 4

**Summary:**

The authors study impartial $k$-selection with predictions, where agents nominate each other and the selection mechanism has access to a predicted set of "best" agents. A selection mechanism is impartial if the selection of an agent is independent of that agent's votes. They say a mechanism is $\alpha$-consistent it it selects a committee with at least an $\alpha$-fraction of the maximal total votes when the prediction matches the maximum vote committee and $\beta$-robust if it always selects a $\beta$-approximate committee regardless of the prediction. The authors extend existing mechanisms for different values of $k$ (uniform permutation, bidirectional permutation, and $k$-partition with permutation) to the setting with predictions, achieving higher consistency-robustness tradeoffs. Through constructions, they also show their tradeoffs are not too far from optimal.

**Questions:**

1. I was a little bit thrown off by the terminology of a prediction being "correct" or "incorrect" based on whether it agrees with the set of nodes in $G$ with maximal indegree. Since this can be checked by the mechanism, it seems to imply that the prediction is useless; as the mechanism always knows what the "correct" prediction would have been. But my understanding is that the existence of a prediction affects the incentives of the voters, allowing different mechanisms to be impartial by using the prediction. Is this understanding correct? If so, I would suggest adding a sentence clarifying the terminology "correct" and "incorrect" and explaining why having a prediction is useful even if its "correctness" is immediately clear.

**Ethical Concerns:**

["NO or VERY MINOR ethics concerns only"]

**Final Justification:**

I had no serious concerns and continue to think this is a strong paper.

**Limitations:**

The authors acknowledge optimality gaps in their results, which are clearly presented. I see no potential negative societal impact of this work.

**Quality:**

4

**Strengths And Weaknesses:**

Strengths:
1. This is a really nicely executed paper, well-grounded in the literature and building on existing results in a novel direction.
2. The quality and clarity are excellent.

Weaknesses:
1. If I had to fish for a weakness, I'd say the audience for this work at NeurIPS may be a little narrow (it's perhaps marginally closer to AAAI/EC/IJCAI), although it's certainly in scope and the discussed applications in AI safety are nice.

Quality: very high. I didn't check the appendix, but the descriptions of the mechanisms make a lot of sense.

Clarity: Figure 1 is *excellent*. Introduction and related work are remarkably clear and thorough. Technical sections are very clear.

Significance: Provides improved bounds for impartial selection in a new setting. The application to multi-agent AI systems is intriguing.

Originality: Adds a new prediction input to the impartial selection setting, which is natural and provides new results.

Minor:
Line 171 definition of $N_S^-$ says it's the cardinality of the set.

---

> ### Author Rebuttal · Authors · 2025-07-29
>
> Thank you for your evaluation, comments, and suggestions. Thanks in particular for pointing out the typo in the definition of $N^-_S$.
>
> Your understanding regarding predictions is correct. In the mechanism design literature (to which we contribute), predictions are powerful since they convey information about the state of the system without being subject to strategic misreporting by the agents.
> In that sense, the use of the words "correct" and "incorrect" (though standard in the literature on algorithms with predictions) is indeed a bit misleading since the latter refers to a case where the behavior of the agents does not align with the prediction of the ML algorithm, but this does not mean that the agents or the ML algorithm were making a mistake.
> To avoid confusion, we will instead use the slightly weaker terms "accurate" and "inaccurate" (as the paper by Xu and Lu we cite).

---

> > ### Comment · Reviewer_jF4z · 2025-08-01
> >
> > That update sounds good. Thanks for the nice submission!

---

### Official Review · Reviewer_S2nN · 2025-07-10

**Clarity:** 3
**Significance:** 3
**Originality:** 3
**Rating:** 4
**Confidence:** 4

**Summary:**

This paper studies the impartial mechanism with prediction in majority voting. The input is given as a graph with vertices referring to the entities and an arc from $j$ to $i$ indicating that $j$ nominates $i$. The $k$ entities with the most vote are selected. The prediction $\hat{S}$ is a set of $k$ entities, also given as an input.

The performance bounds are measured by consistency and robustness. The consistency is prefermance of the proposed mechanism when the prediction is accurate, while the robustness is that when the prediction is wrong. The authors focus on impartial mechanisms and present the following main results. Below, I use $(\alpha,\beta)$ to represent $\alpha$-consistent and $\beta$-robust.

1. when $k=1$, for any $\rho \in [1/2,1]$, a randomized mechanism achieves $(\rho,1-\rho)$;
2. when $k=2$, a deterministic mechanism achieves $(1,1/2)$, and a randomized mechanism achieves $(\frac{2}{3}+\frac{1}{3}\rho, \frac{2}{3}- \frac{1}{6}\rho)$ for all $\rho\in [0,1]$;
3. when $k\geq 3$, a deterministic mechanism achieves $(1,\frac{1}{k})$, and a randomized mechanism that $\alpha \geq 1-\frac{1}{2k}$, which lossing $O(\frac{1}{n})$ of $\beta$ when compared to the best-known mechanism without prediction.

The authors also present upper bounds of $(\alpha, \beta)$. Result 1 achieves the best possible, and moreover, the gaps between upper and lower bounds of other settings are not large.

**Questions:**

Q1 : For plurality voting, you only discussed the randomized mechanisms without prediction and designed randomized mechanisms with predictions. Is this because the impossibility result for deterministic mechanisms mentioned earlier (line 199) also extends to the plurality voting setting?

**Ethical Concerns:**

["NO or VERY MINOR ethics concerns only"]

**Final Justification:**

This paper has no serious concerns and can be accepted.

**Limitations:**

Yes

**Paper Formatting Concerns:**

No formatting concerns

**Quality:**

3

**Strengths And Weaknesses:**

The research question is interesting. Mechanism design with prediction has recently become an active research direction, particularly for those problems with strong impossibility results. The core idea is to use predictions to improve the algorithm's performance when the predictions are accurate (consistency), while maintaining (or nearly maintaining) the best-known bounds when the predictions are incorrect (robustness). This paper introduces the framework prediction to the impartial mechanisms in majority voting.

The results are non-trivial. The maniscript considers several settings, depending on $k$ and the graph structure, and for each of them, it presents upper and lower bounds. Although the presented bounds are not tight, the gap between them is relatively small, which I find acceptable.

The paper is well-written in general. The reserach question is well motivated and the results are nicely summarized in Figure 1.

For the techniques, most of the presented algorithms are established on the existing ones (without prediction). The presented algrithms share the similar ideas of the existing ones or use them as subrotines, though with adaptations to handle the prediction setting. Although these adaptations are not trivial, they are not particularly difficult to devise.

Other minor comments:

line 261, Theorem 3.3. From the proof, it seems that \delta \geq 2 is required, which is not mentioned in the statement. For the current statement, if \delta=1, then \beta(1)=0. You may want to rephrase the statement.

line 256, show -> have shown

line 198, using $\Delta(G)$ for $\Delta_1(G)$ has been mentioned in line 174

---

> ### Author Rebuttal · Authors · 2025-07-29
>
> Thank you for your detailed comments and the questions you have raised.
> We will happily fix any minor issues.
> In particular, the assumption $\Delta\geq 2$ in Theorem 3.3 is indeed relevant: If $\Delta=1$, all vertices have indegree $1$ and any mechanism is trivially $1$-robust.
>
> Regarding the question about deterministic mechanisms for plurality voting, the fact that no mechanism can achieve constant robustness indeed holds in this setting as well, due to an impossibility result of Holzman and Moulin (Econometrica, 2013).
> This is why we restrict ourselves to randomized mechanisms in the whole section.
> We appreciate the comment, as we think it is indeed valuable to point this out more clearly when introducing plurality voting.

---

> > ### Comment · Reviewer_S2nN · 2025-08-04
> >
> > Thank you for the response that addresses my question.

---

### Decision · Program_Chairs · 2025-09-17

**Decision:**

Accept (poster)

**Comment:**

The authors study impartial selection (where agents mutually nominate each other) in the presence of predictions about the set of agents who will be nominated. In line with prior work in the predictions setting, they study the optimal tradeoff between consistency and robustness and present upper and lower bounds in a variety of settings (selecting 1, 2, and \ge 3 vertices).

Strengths:
- Reviewers praised the writing quality, exposition, illustrations (particularly Figure 1!), and motivation of the work. The paper is quite polished overall.
- The technical results are nontrivial, although they build on prior work. Some reviewers noted that the technical results are a bit incremental (mkRF).

Weaknesses:
- The authors leave some gaps between their upper and lower bounds for most settings. While this is perhaps to be expected from a paper that initiates the study of a new setting, it is still a shortcoming of the paper.
- The paper lacks empirical analysis (mkRF) and has a perhaps narrow scope (jF4z).